# MaVEn: An Effective Multi-granularity Hybrid Visual Encoding Framework for Multimodal Large Language Model

**Chaoya Jiang**[1][†], **Hongrui Jia**[1][†], **Haiyang Xu**[2][*], **Wei Ye**[1][*], **Mengfan Dong**[1],
**Ming Yan**[2], **Ji Zhang**[2], **Fei Huang**[2], **Shikun Zhang**[1]
[1] National Engineering Research Center for Software Engineering, Peking University
[2] Alibaba Group
{jiangchaoya, wye, zhangsk}@pku.edu.cn,
{shuofeng.xhy, fei.huang}@alibaba-inc.com

## Abstract

This paper presents MaVEn, an innovative Multi-granularity Visual Encoding framework designed to enhance the capabilities of Multimodal Large Language Models (MLLMs) in multi-image reasoning. Current MLLMs primarily focus on single-image visual understanding, limiting their ability to interpret and integrate information across multiple images. MaVEn addresses this limitation by combining discrete visual symbol sequences, which abstract coarse-grained semantic concepts, with traditional continuous representation sequences that model fine-grained features. This dual approach bridges the semantic gap between visual and textual data, thereby improving the model's ability to process and interpret information from multiple images effectively. Additionally, we design a dynamic reduction mechanism by for long-sequence continuous features to enhance multi-image processing efficiency. Experimental results demonstrate that MaVEn significantly enhances MLLMs' understanding in complex multi-image scenarios, while also improving performance in single-image contexts.

## 1 Introduction

Current multimodal large models (MLLMs) [37] concentrate on understanding single images [24, 43, 36, 22], which significantly restricts their ability to interpret and integrate information across multiple images. As shown in Figure 1, typical scenarios [17] involving multiple images include Knowledge Based VQA, Visual Relation Inference, Multi-image Reasoning and so on. These scenarios present a wide array of practical applications.

Present strategies predominantly adopt a data-centric approach, where methods such as those proposed in [1, 15, 41, 2] aim to strengthen the multi-image capabilities of Multimodal Large Language Models (MLLMs) by introducing interleaved image-text data during the pre-training and fine-tuning phases. Although some efficacy has been achieved, training solely based on interleaved data still falls short in many multi-image scenarios. This is primarily because current MLLMs remain fundamentally designed for single-image scenarios. This raises the question of whether the visual feature encoding and bridging methods of MLLMs, originally designed for single-image input scenarios, are suitable for multi-image inputs.

Current MLLMs encode visual inputs using either discrete symbol encoding [11, 13, 35, 4] or continuous sequence encoding [7, 27, 34]. For the continuous sequence feature encoding category, the

---

[*]corresponding authors.
[†]These authors contributed equally to this work.

38th Conference on Neural Information Processing Systems (NeurIPS 2024).

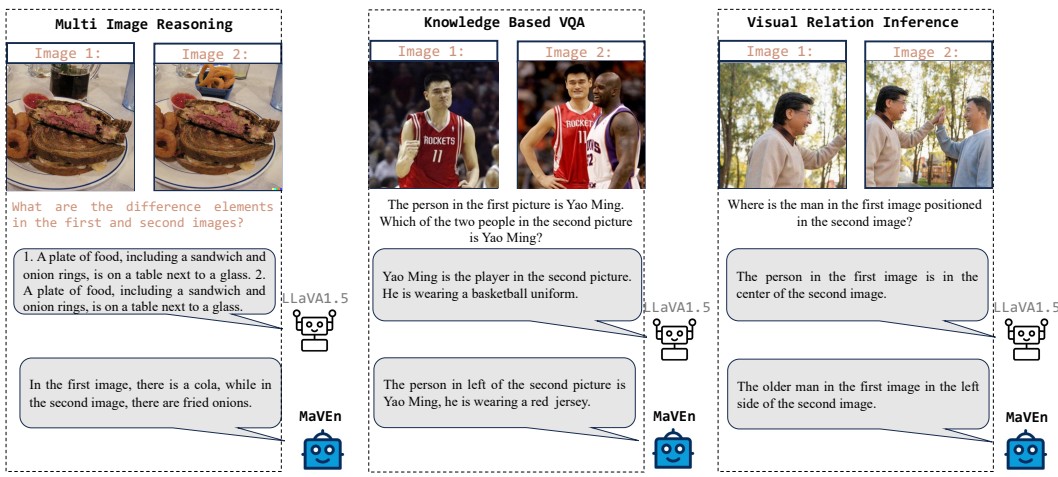

Figure 1: We compared the performance of the classic single-image task trained MLLM LLaVA1.5 [22] and our model in three multi-image scenarios including Multi Image Reasoning, Knowledge Based VQA and Visual Relation Inference. LLaVA1.5 exhibits significant limitations in multi-image scenarios.

following issues are present: (1) Excessively Long Visual Feature Sequences: Most current MLLMs utilize linear layers to bridge the visual sequence outputs from Vision Transformers (ViTs) [7]. Given the lengthy encoding sequences of images and the finite context input length of current MLLMs, the extended feature sequence inputs in multi-image contexts result in complex computational overhead and adversely affect model performance. (2) Imprecise Visual Information Encoding: Some MLLMs employ fixed-length latent queries to encode visual features through architectures like Q-Former [18]. While this approach somewhat reduces the length of visual sequences, recent studies [20] suggest that it still does not encode visual information from images with sufficient accuracy. There remains a misalignment with textual representations, leading to the model's confusion regarding visual information.

Moreover, recent works [11, 13, 4] have started to explore encoding images as discrete symbol sequences. Compared to complex continuous visual representations, discrete visual representations offer simpler and clearer high-level semantic abstractions, closely aligning with the discrete nature of textual representations. Consequently, discrete visual encoding looks like more conducive to complex multi-image reasoning. However, given that discrete visual representations tend to be coarser in granularity, relying solely on them may overlook fine-grained details within images [40].

In this study, we introduce **MaVEn**: an effective and efficient **M**ulti-gr**a**nularity Hybrid **V**isual **E**ncoding framework. MaVEn utilizes discrete visual symbol sequences to abstract coarse-grained semantic concepts, aiding in multi-image understanding, while traditional continuous representation sequences model fine-grained features to support detailed understanding. Accordingly, we investigate the synergy of multi-granularity visual features within the novel framework, design a dynamic reduction mechanism for long-sequence continuous features to enhance multi-image processing efficiency, and propose a multi-stage model training methodology aimed at improving multi-image comprehension. Experimental results demonstrate that the proposed method effectively enhances the understanding capabilities of MLLMs in complex multi-image scenarios, while also improving performance in single-image contexts. In summary, the contributions of this study include:

- We introduce a framework that combines discrete and continuous visual representations to enhance multi-image reasoning in MLLMs. This framework improves the model's ability to process and interpret information from multiple images effectively.

- We design a dynamic reduction mechanism for long-sequence continuous visual features to increase the efficiency of multi-image processing in MLLMs.

- Our approach demonstrates remarkable performance across various multi-image scenarios and also shows advantages in standard single-image benchmarks.

## 2 Related Work

### 2.1 Multimodal Large Language Models

Existing Multimodal Large Language Models (MLLMs) [37] typically consist of a visual encoder, a visual interface, and a large language model (LLM). The visual encoder converts visual data into continuous sequence features. The visual interface then maps these features into the LLM's semantic space, allowing the LLM to process visual information. Current research focuses on developing effective visual interfaces. There are two main types: Latent-Query based Models: Used in MLLMs like BLIP-2 [18] and MiniGPT-4 [43], this approach uses a fixed number of learnable latent vectors as query vectors in an attention mechanism. These vectors interact with visual sequence representations to summarize and integrate visual information, effectively reducing sequence length but potentially losing some visual details. Linear Mapping Models: Used in MLLMs like LLaVA [24], this method directly maps visual feature sequences into the LLM's text embedding space via a linear layer. This approach retains complete visual information but results in longer output sequences.

### 2.2 Visual Semantic Encoding Representations in MLLMs

Efficient visual semantic encoding has become a key research area for MLLMs. Researchers have developed various methods to represent visual information, including: Continuous Visual Encoders: Examples include Visual Transformer (ViT) [7] and Swin-Transformer [27]. ViT segments images into patches and processes them sequentially, while Swin-Transformer uses a sliding window mechanism to capture local structures more efficiently. These methods excel in capturing image details but face challenges in aligning with textual encoding [20]. Discrete Visual Encoders: These methods encode images into discrete sequences similar to text tokens, aligning visual and textual information more closely. Examples include VQ-VAE [35], VQ-GAN [13], and SEED [11]. VQ-VAE uses self-supervised learning to create a visual vocabulary from image patches. VQ-GAN combines VQ-VAE with generative adversarial networks to capture semantic information and generate high-quality images. SEED, the latest approach, encodes images into discrete visual sequences with one-dimensional causal dependencies, aiming to extract high-level semantics for visual understanding and generation tasks.

## 3 Method

As illustrated in Figure 2, we proposes an MLLM architecture that leverages multi-granularity visual features for enhanced multi-image understanding. Visual images are encoded as both discrete symbol sequences and continuous high-dimensional vector sequences. The discrete visual symbol sequences capture essential coarse-grained visual concepts from the images, while the continuous vector sequences encapsulate fine-grained details. Furthermore, to minimize redundant and irrelevant visual representations in the continuous visual sequences and thereby reduce the input context length in multi-image scenarios, we also introduces a dynamic reduction strategy for visual features, guided by textual semantics.

### 3.1 Multi-Granularity Hybrid Encoding

As shown in Figure 2 (a), assume the input to the MLLM is $\{S, T\}$, where $S = \{I_1, I_2, \ldots, I_K\}$ represents a collection of $K$ images, and $T$ denotes the corresponding textual content. For each image $I_k$, $k \in \{1, 2, \ldots, K\}$, we employ both the discrete visual encoder SEED [11] and the continuous visual encoder ViT [7] for encoding.

**Visual Continuous Encoding**: we utilize the Vision Transformer (ViT) model, which is widely adopted by most modern Multimodal Large Language Models (MLLMs). For an RGB image $I_k$ with dimensions $W \times H \times 3$, the image is partitioned into patches of size $p \times p$, resulting in $\frac{W}{p} \times \frac{H}{p}$ patches. These patches are then encoded by the ViT visual encoder into a continuous visual sequence: $V_c^k = [\vec{v}_1^k, \vec{v}_2^k, \ldots, \vec{v}_{n_c}^k]$. Here, $n_c = \frac{W}{p} \times \frac{H}{p}$, and $\vec{v}_i^k \in \mathbb{R}^z$ represents a continuous vector of $z$ dimensions. Subsequently, we utilize the text-semantics-aware patch reduction module (details of which will be elaborated in Subsection 3.2) to select patch features relevant to the input textual content $T$, thereby reducing the sequence length of $V_c^k$, while preserving essential fine-grained information. The reduced feature sequence is denoted as $V_c^k = [\vec{v}_{p_1}^k, \vec{v}_{p_2}^k, \ldots, \vec{v}_{p_{m_c}}^k]$, $m_c \ll n_c$. Finally, we utilize

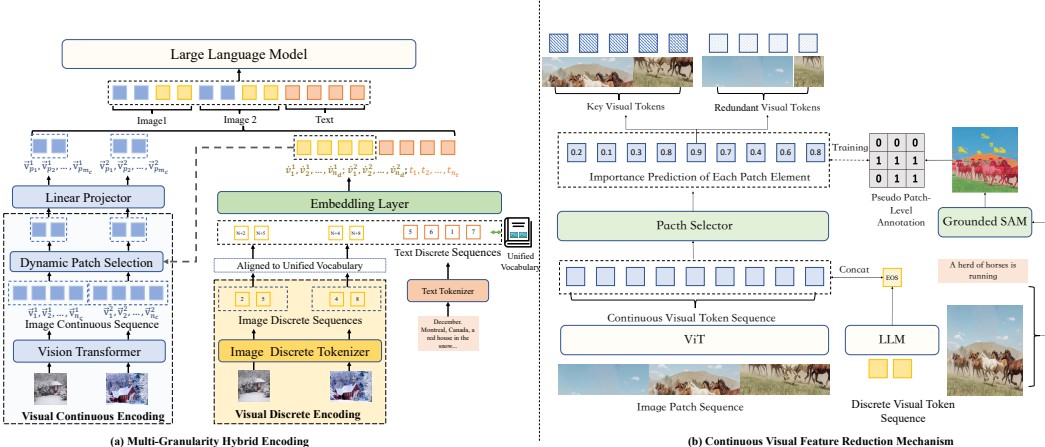

**Figure 2:** Subfigure (a) illustrates the structural schematic of our proposed Multi-Granularity Hybrid Encoding, while subfigure (b) demonstrates the mechanism for the reduction of continuous visual tokens under the guidance of discrete visual information.

an Multi-Layer Perceptron, akin to that used in LLaVA 1.5, as a bridging projector to project $V_c$ into the semantic space of the LLM embedding layer.

**Visual Discrete Encoding**: image $I_k$ is tokenized by the image discrete tokenizer $\mathbf{D}_v$ (for details on the design and training of the discrete tokenizer, please refer to the original work [11]) into a visual discrete symbol sequence $V_d = [d_1^k, d_2^k, \ldots, d_{n_d}^k]$, where $d_i^k \in [1, 2, \ldots, \mathsf{N}_v]$. Here, $\mathsf{N}_v$ denotes the size of the visual discrete encoding vocabulary.

**Unified Multimodal Vocabulary:** Given that text modalities naturally possess a discrete vocabulary, merging the visual discrete vocabulary with the textual discrete vocabulary forms a unified multimodal vocabulary. The advantage of this approach lies in its ability to achieve a unified representation of both visual and textual modalities, effectively addressing the semantic gap between them. Assume that the vocabulary size of the LLM is $N$, and the vocabulary size of the visual discrete tokenizer is $N_v$. The expanded multi-modal unified vocabulary size thus becomes $N_u = N + N_v$. Concurrently, we align each element in $V_d$ with the index of the unified vocabulary to obtain the final discrete encoding: $V_d = [\hat{v}_1^k, \hat{v}_2^k, \ldots, \hat{v}_{n_d}^k]$, where $\hat{v}_i^k = d_i^k + N$. Finally, the weight matrix of the LLM's embedding layer, $W$, is also expanded from $N \times z$ to $N_u \times z$. Consequently, the weight matrix of the LLM's embedding layer, $W$, is expanded from $N \times z$ to $N_u \times z$. This adjustment enables the embedding layer of the LLM to concurrently encode features from both visual and textual discrete tokens. The embedded representation of $V_d$ is denoted as $[\dot{v}_1^k, \dot{v}_2^k, \ldots, \dot{v}_{n_d}^k]$ which is output by the expanded embedding layer. Finally, we sequentially insert the continuous visual tokens before the discrete visual token embeddings outputted by the embedding layer. The final visual representation inputted into the LLM is: $[\vec{v}_{p_1}^k, \vec{v}_{p_2}^k, \ldots, \vec{v}_{p_{m_c}}^k, \dot{v}_1^k, \dot{v}_2^k, \ldots, \dot{v}_{n_d}^k]$.

## 3.2 Continuous Visual Tokens Reduction Mechanism

We aim for the discrete visual tokens to abstract high-level, coarse-grained semantics from the images, while the continuous visual tokens complement this with low-level, fine-grained details. However, we found that the continuous visual tokens output by the Vision Transformer (ViT) encompass a considerable amount of redundancy, with many tokens possessing repetitive or superfluous semantics. Consequently, as shown in Figure 2 (b), we propose a continuous visual token reduction mechanism guided by the coarse-grained semantics of discrete visual tokens, aimed at achieving semantic synergy between coarse-grained and fine-grained representations.

Firstly, after obtaining the sequence of discrete visual tokens $V_d = [\hat{v}_1^k, \hat{v}_2^k, \ldots, \hat{v}_{n_d}^k]$, we append an <EOS> token to it. This sequence $V_d = [\hat{v}_1^k, \hat{v}_2^k, \ldots, \hat{v}_{n_d}^k, t_{eos}]$ is then passed through the LLM to obtain the final layer's output hidden state of the <EOS> token denoted as $h_{eos}$, which represents the global information of the discrete visual tokens: $h_{eos} = \mathrm{LLM}([\hat{v}_1^k, \hat{v}_2^k, \ldots, \hat{v}_{n_d}^k, t_{eos}])$. we then concatenate the EOS token with each image patch token as $\dot{\vec{v}}_i^k = concat(\vec{v}_i^k, h_{eos})$, where $\vec{v}_i^k \in R^z, h_{eos} \in R^z, \dot{\vec{v}}_i^k \in R^{2z}, i \in \{1, 2, \ldots, n_c\}$. Then the concatenated patch features $\dot{\vec{v}}_i^k$ are

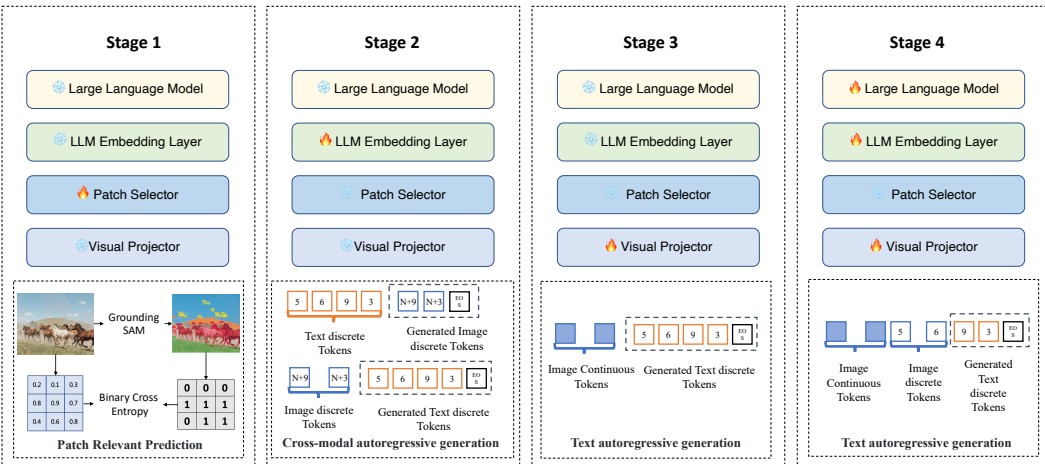

Figure 3: The diagram illustrates the training schematic for MaVEn. We divide the training of MaVEn into four stages, where the snowflake icon indicates that the model parameters are frozen during training, and the flame icon indicates that the model parameters are updated during training.

fed to the patch selector. The patch selector is an Multilayer Perceptron (MLP) denoted as $\mathbf{F}$ that contains three linear layers and is used to predict the relevant score between patches and the discrete visual tokens. The output of the last linear layer has only one dimension and will be fed to a Sigmoid activation function to predict the relevant score $a_i$ with the discrete visual tokens as $a_i = \mathbf{F}(\vec{v}_i^k, h_{eos})$ According to the prediction of patch selector $\mathbf{F}$, the top-m key image patch tokens are kept and the unselected patch tokens which generally have lower relevant scores will be discarded. Finally, we reconstruct the reduced visual sequence as $v^k = \left[\vec{v}_{p_1}^k, \cdots, \vec{v}_{p_m}^k\right]$, where $m = n_c \times \alpha$, and $\alpha$ is a hyper-parameter and named Keeping Ratio which is used to control the proportion of selected patches to total patches.

**Construction of Patch-level pseudo-label annotation:** To train the patch selector, we constructed patch-level pseudo-label annotations based on Grounding SAM [32] (a recent state-of-the-art open-text vocabulary semantic segmentation model). We observed that the high-dimensional semantics encapsulated within the discrete visual tokens are largely consistent with the semantics of the image captions. Inspired by this observation, as shown in Figure 2 (b), we opted to use image captions as a proxy for the high-dimensional semantic abstraction of the image. We employed Grounding SAM to perform text-guided semantic segmentation of the images. After obtaining the semantic segmentation pixel masks, we computed the overlap between each patch and the mask labels. If there is an overlap, the corresponding patch label is set to 1; otherwise, the label is set to 0.

### 3.3 Training Paradigm of MaVEn

The training process of MaVEn is divided into four stages. In the first stage, we utilized image-text datasets like COCO [21] and Visual Genome (VG) [14] to annotate 1 million semantic segmentation masks with textual annotations, based on Grounding SAM [32]. These masks were subsequently converted into patch-level pseudo-labels. Utilizing this dataset, we trained the Patch Selector while keeping other model parameters frozen.

In the second stage, we exclusively trained the embedding layer of the LLM (Large Language Model) to adapt to our expanded vocabulary for the LLM. Consequently, we utilized the LLaVA 558k single-image pretraining dataset [24] and the MMC4 interleaved image-text dataset [45] for training. At this stage, we employed only the visual discrete encoding, eschewing the visual continuous encoding, with the aim of adapting the LLM embedding layer to the expanded unified vocabulary. We trained using a cross-modal autoregressive generation task; given an input that might contain images, we obtained tokenized discrete sequences through the text and image tokenizers. This enabled us to generate discrete image token sequences from text discrete token sequences and vice versa.

In the third stage, our objective is to optimize the visual projector so that the continuous visual tokens, after being processed by the visual projector, align with the semantic space distribution of the unified multimodal vocabulary embeddings. Therefore, during this phase, we train solely the visual projector.

| Method | Multi Modal Dialogue | Visual Story Telling List | Visual Relation Inference | Multi Modal Cloze | Knowledge Grounded QA | Text Rich Images QA | Multi Image Reasoning |
|---|---|---|---|---|---|---|---|
| BLIP-2 [19] | 11.96 | 20.10 | 3.67 | 18.25 | 39.73 | 30.53 | 39.53 |
| mPLUG-Owl [36] | 12.67 | 19.33 | 5.40 | 16.25 | 33.27 | 32.47 | 42.50 |
| InstructBLIP [6] | 33.58 | 24.41 | 11.49 | 21.20 | 47.40 | 44.40 | 48.55 |
| LLaMA-Adapter-v2 [10] | 14.22 | 17.57 | 13.51 | 18.00 | 44.80 | 32.00 | 44.03 |
| LLaVA [25] | 7.79 | 10.70 | 8.27 | 15.85 | 36.20 | 28.33 | 41.53 |
| MiniGPT-4 [44] | 13.70 | 17.07 | 7.95 | 16.60 | 30.27 | 26.40 | 43.50 |
| LLaVA-1.5 [23] | 27.17 | 14.32 | 11.62 | 31.65 | 46.4 | 38.87 | 44.58 |
| Otter [15] | 15.37 | 15.57 | 11.39 | 16.00 | 41.67 | 27.73 | 43.85 |
| OpenFliamingo [2] | 16.88 | 24.22 | 13.85 | 21.65 | 32.00 | 30.60 | 41.63 |
| VPG-C [17] | **37.50** | **25.20** | 25.90 | 22.15 | 48.60 | 44.93 | 50.28 |
| MaVEn | 34.63 | 21.53 | **30.24** | **33.35** | **51.53** | **47.33** | **54.38** |

Table 1: Average results of **zero-shot evaluation** on each task of **DEMON Benchmark** [17].

| Method | Vision Encoder | Language Model | Avg. All | Avg. Img | Avg. Video |
|---|---|---|---|---|---|
| BLIP-2 [19] | ViT-g (1.3B) | Vicuna (7B) | 46.4 | 49.7 | 36.7 |
| mPLUG-Owl [36] | ViT-L (0.3B) | LLaMA (7B) | 34 | 37.9 | 23 |
| InstructBLIP [6] | ViT-g (1.3B) | Vicuna (7B) | 53.4 | 58.8 | 38.1 |
| LLaMA-Adapter-v2 [10] | ViT-L (0.3B) | LLaMA (7B) | 32.7 | 35.2 | 25.8 |
| Otter [15] | ViT-L (0.3B) | LLaMA (7B) | 33.9 | 35.2 | 30.4 |
| LLaVA [25] | ViT-L (0.3B) | Vicuna (7B) | 33.5 | 37.0 | 23.8 |
| MiniGPT-4 [44] | ViT-g (1.3B) | Vicuna (7B) | 42.8 | 47.4 | 29.9 |
| LLaVA-1.5 [23] | ViT-L | Vicuna (7B) | 58.6 | 66.1 | 37.3 |
| MaVEn | ViT-L + SEED (1.3B) | Vicuna (7B) | 60.89 | 65.85 | 42.11 |

Table 2: Average results of **zero-shot evaluation** on each task category of **SEED Benchmark** [17].

We train using the LLaVA 558K image-text caption dataset, where images are encoded solely as sequences of continuous visual tokens: $V_c^k = [\vec{v}_{p_1}^k, \vec{v}_{p_2}^k, \ldots, \vec{v}_{p_{m_c}}^k]$ without employing visual discrete coding. The model is required to generate captions for the images based on the visual input.

In the fourth stage, we introduce instruction fine-tuning data with the aim of enhancing the MLLM's capability to follow human instructions. During this phase, the MLLM undergoes comprehensive fine-tuning with the LLaVA 665k instruction fine-tuning datasets, unfreezing all model parameters except for those of the visual encoder and patch selector for training and optimization.

# 4 Experiments

## 4.1 Experiment Setting

**Dataset:** We initially generated 1 million pseudo-labels for patch-level text semantic relevance by utilizing the COCO[21], Visual Genome (VG)[14], and RefCOCO datasets [38], following the methodology delineated in Subsection 3.2 and leveraging Ground SAM. These pseudo-labels were subsequently employed to train the patch selector within MaVEn. During the second phase of model training, the embedding layer of MaVEn was refined using the MMC4-core dataset [45] and the LLaVA 558K single-image pre-training dataset [22]. In the third phase, the visual projector component of MaVEn was further trained using the LLaVA 558K single-image dataset. Finally, in the final phase, we fine-tuned the model with the LLaVA 665K instruction fine-tuning dataset.

**Training Settings:** MaVEn utilizes the ViT-L model [31] with a patch size of $14 \times 14$ and is pre-trained at a resolution of $336 \times 336$, resulting in a continuous token length of 567 for the encoded image. For image discrete tokenization, SEED [11] is employed to tokenize the image into 32 discrete tokens. For the continuous visual tokens, during patch reduction, we set the Keeping Ratio to **0.25**, meaning that only 25% of the continuous tokens are retained. Consequently, the length of the final continuous visual token sequence decreases from 576 to 144, while the length of the discrete token sequence is 32. Ultimately, the entire visual hybrid encoding sequence has a length of 176. The large language model Vicuna [42], with 7 billion parameters, is used to handle multi-modal features. The AdamW optimizer [28] is used for optimization. During the instruction tuning stage, the entire model is trained for 1 epoch with a learning rate of 2e-5 and a batch size of 256. All experiments was performed using 8 NVIDIA A100 GPUs, each with 80GB of memory.

| Method | #Params | General VQA | | General VQA (Zero-shot) | | | Zero-shot Multi-modal Benchmarks | | |
|---|---|---|---|---|---|---|---|---|---|
| | | VQAv2 | GQA | VizWizQA | TextVQA | SciQA | MME | MMBench | MM-Vet |
| BLIP-2 [19] | 8.2B | 65.0 | 41.0 | 19.6 | 42.5 | 61.0 | 1293.84 | - | 22.4 |
| InstructBLIP [6] | 8.2B | - | 49.2 | 34.5 | 50.1$^\dagger$ | 60.5 | 1212.82 | 36.0 | 26.2 |
| Unified-IO$_{XL}$ [29] | 2.9B | 77.9 | - | 57.4$^\ddagger$ | - | - | - | - | - |
| PaLM-E-12B [8] | 12B | 76.2 | - | - | - | - | - | - | - |
| Shikra [5] | 7.2B | 77.4 | - | - | - | - | - | 58.8 | - |
| Qwen-VL-Chat [3] | 9.6B | 78.2 | 57.5 | 38.9 | 61.5$^\ddagger$ | **68.2** | 1487.58 | 60.6 | - |
| LLaVA [23] | 7.2B | 71.3 | 41.3 | 36.7 | 50.2$^\dagger$ | 61.5 | 502.82 | 36.2 | 28.1 |
| MiniGPT-4 [23] | 7.2B | 65.2 | 30.8 | 30.2 | 52.3$^\dagger$ | 58.4 | 581.67 | 23.0 | 22.1 |
| LLaVA1.5 [23] | 7.2B | 78.5 | 62.0 | 50.0 | 58.2$^\dagger$ | 66.8 | 1510.70 | 64.3 | **30.5** |
| MaVEn | 7.2B | **79.1** | **62.5** | **50.5** | 59.8$^\dagger$ | 67.3 | **1530.10** | **65.2** | 30.4 |

Table 3: **Performance comparison on visual question answering and zero-shot multi-modal benchmarks.** For VQA, accuracy is reported. Note that specialists are fine-tuned on each individual dataset. † denotes OCR inputs are utilized. ‡ indicates the model has trained on the dataset.

## 4.2 Main Results

To validate the effectiveness of MaVEn in multi-image scenarios, we evaluated MaVEn's performance multi-image visual understanding and reasoning. Within the multi-image visual understanding context, we assessed the model using DemonBench[17] and SEED-Bench[16]. DemonBench comprises seven scenarios involving multi-image reasoning and understanding, including tasks such as Multi-Modal Dialogue, Visual Relation Inference, Knowledge Grounded QA, and Multi-Image Reasoning. SEED-Bench, on the other hand, encompasses questions related to video comprehension. Additionally, we also tested the performance of MaVEn in single-image scenarios.

### 4.2.1 Effectiveness of MaVEn on Multi-image Visual Comprehension

**Results on DemonBench**: As shown in the Table 1, we evaluated our model on DemonBench, comparing it with several multi-image data-trained MLLM models such as Openflamingo, Otter, VPG-C, as well as single-image scenario MLLM models. Our model attained the highest scores in tasks such as Visual Relation Inference, Multi-Modal Cloze, Text-Rich Images QA, Knowledge-Grounded QA, and Multi-Image Reasoning, underscoring MaVEn 's significant superiority in multi-image understanding and reasoning tasks. It also achieved comparable performance in Visual Storytelling and Multi-Modal Dialogue.

**Results on SEED-Bench**: Furthermore, we assessed our model on SEED-Bench [16], particularly focusing on video understanding tasks like action prediction, action Recognition and procedure understanding. As shown in Table 2 The experiments revealed that our approach significantly outperformed existing models like LLaVA 1.5, Otter and so on. For instance, MaVEn exhibited a 12-point improvement over Otter [15] in video understanding (30.4 -> 42.11). These findings underscore MaVEn 's effectiveness in multi-image understanding scenarios.

### 4.2.2 Effectiveness of MaVEn on Single-image Visual Comprehension

We intend to explore the influence of MaVEn on the model's abilities of single image visual comprehension and generation. To achieve this objective, we carried out assessments on common benchmarks, such as Visual Question Answering (VQA) [12, 30, 33] and recently designed MLLM-focused Multi-modal Benchmarks including MME [9], MMBench [26], MM-Vet [39].

**Results on Benchmark Tasks** As summarized in Table 3. We compared performance of MaVEn to other SOTA MLLMs such as BLIP2[19], InstructBLIP [6], Shikra [5], and Qwen-VL-Chat [3]. Our experimental results show that our approach can also successfully enhances the performance across a range of single image understanding task. Notably, MaVEn outperforms LLaVA-1.5 [25] in terms of consistency and accuracy across all VQA datasets.

**MLLM-oriented Multi-modal Benchmarks.** We also evaluate MaVEn on four recently popular single-image multi-modal benchmarks in a zero-shot manner. The results of our evaluation are listed in Table 3. We discovered that after implementing MaVEn, all three models exhibited improvements

| Discrete | Continuous | SEED-Bench | DEMONBench | VQA | MMBench |
|:---:|:---:|:---:|:---:|:---:|:---:|
| ✓ | ✗ | 43.2 | 24.19 | 56.07 | 34.5 |
| ✗ | ✓ | 58.6 | 30.66 | 78.5 | 64.3 |
| ✓ | ✓ | 60.89 | 39.51 | 79.1 | 65.21 |

Table 4: **Ablation evaluation on multi-modal benchmarks** We evaluated the performance of various ablation targets on both multi-image (SEED-Bench, DEMONBench ) and single-image (VQA, MMBench) benchmarks. MMBench [26].

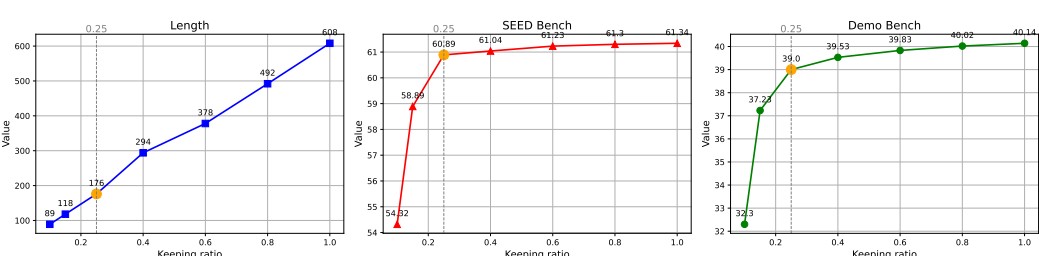

Figure 4: Evaluation Results of MaVEn on different benchmarks with varying Keeping Ratios.

across multiple benchmarks. Notably, for LLaVA and MiniGPT-4, the enhancement was particularly evident on the MME [9] benchmark. For instance, after implementing MaVEn, LLaVA's MME score improved from 581.67 to 653.94. These results highlight MaVEn role in advancing the state-of-the-art in both single-image and multi-image visual comprehension tasks.

## 4.3 Ablation Study

### 4.3.1 Effectiveness of Multi-Granularity Hybrid Encoding

To ascertain the efficacy of multi-granularity hybrid encoding, we conducted training using solely visual discrete encoding and visual continuous encoding, respectively. We then evaluated and compared the outcomes on both multi-image and single-image evaluation benchmarks. The results are detailed in the Table 4 below. We observed that, compared with utilizing only visual continuous encoding or employing a hybrid of visual discrete and continuous encoding, the model solely on visual discrete encoding exhibits subpar performance in both multi-image and single-image contexts. This underperformance is likely due to the nature of discrete visual feature encoding, which, while capturing the high-dimensional information of the image, forfeits a significant amount of low-dimensional, fine-grained details, resulting in a lossy encoding process. As a result, it fares poorly in tasks like image reasoning and understanding, which demand meticulously detailed information. Moreover, the model with only continuous encoding also do not deliver optimal performance, particularly in multi-image tasks. This further indicates that models based solely on visual continuous encoding are unsuitable for multi-image scenarios.

### 4.3.2 Efficient of Continuous Visual Token Reduction Mechanism

To verify the effectiveness of patch reduction, we compared the length of visual tokens at different Keeping Ratios and analyzed the performance across various benchmarks. We experimented with the Keeping Ratios from 0.1 to 1.0. The experimental results are shown in the Figure 4. We observed that when the Keeping Ratio was 0.1, the number of visual tokens decreased to 89, a significant reduction from the initial number of patches. However, the model's performance across multiple benchmarks also significantly declined. Therefore, despite the reduction in the number of visual tokens, the performance loss was too substantial, making it an unsuitable final choice. When the Keeping Ratio was 0.25, the number of visual tokens remained relatively low, but the model's performance was more stable. In this case, the model exhibited balanced performance across various benchmarks, effectively reducing the number of visual tokens while maintaining a high-performance level. Therefore, we ultimately chose a Keeping Ratio of 0.25.

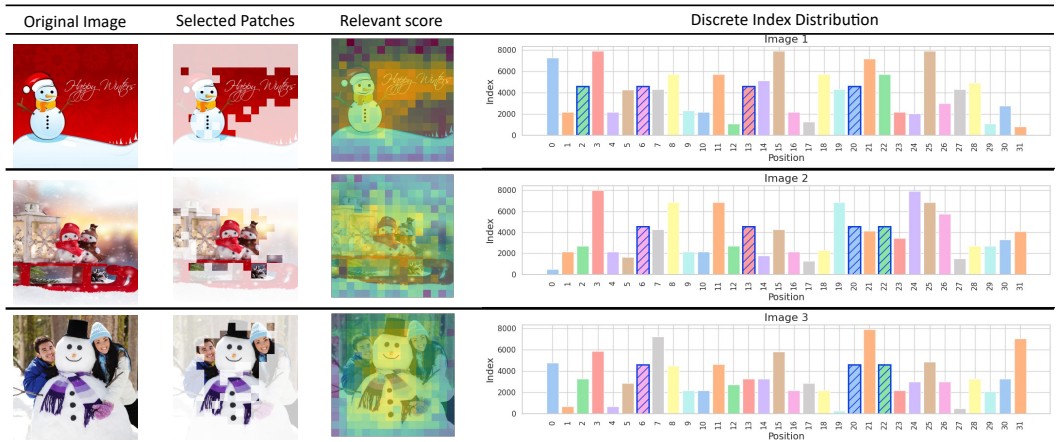

Figure 5: This figure visualizes the distribution of discrete tokens in an image containing index 4568 discrete tokens, along with the relevant score computed based on the Patch Selector and the patches chosen according to the relevant score that are most semantically related to the discrete visual tokens.

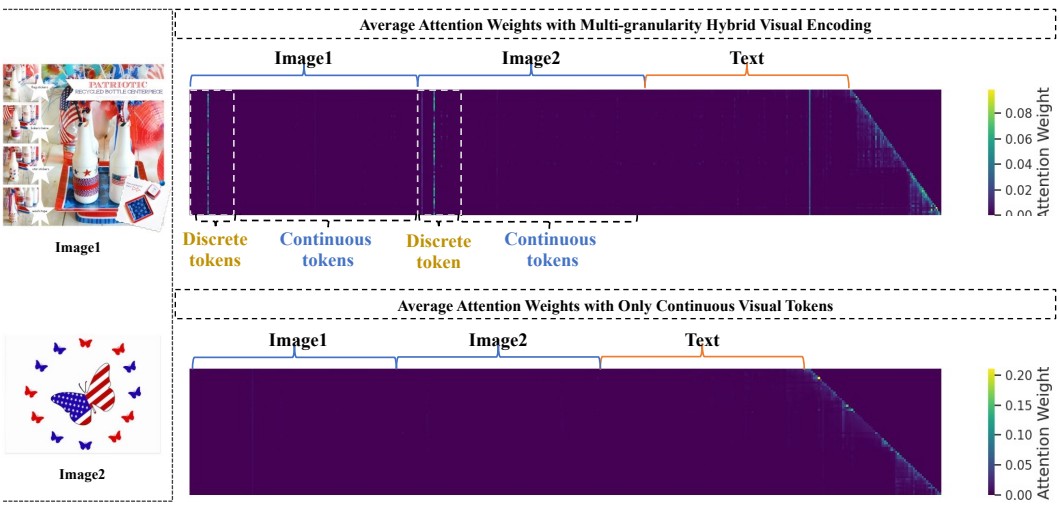

Figure 6: Visualization of the attention maps with and without the visual discrete tokens . We demonstrate the attention maps for the 31-st layers, where the range of visual tokens is indicated by orange and the range of text tokens is indicated by blue.

## 4.4 Qualitative Analysis

**Semantic Granularity of discrete and continuous visual tokens**: To investigate the semantics of discrete tokens, we randomly selected an index from the visual discrete dictionary and searched the CC/SUB dataset for images containing this index in their encoding. As illustrated in Figure 5, we randomly chose three images with index = 4568. We discovered that all three images featured the depiction of a snowman, suggesting that index = 4568 can represent high-level semantics such as a snowman or white snow. We provided the distribution of discrete tokens for these three images and observed that the position of index = 4568 in the discrete sequences was also notably consistent.

**Semantic synergy between visual discrete and continuous representations**: Furthermore, as shown in Figure 5, we also visualize the relevant score between the patches and the semantics of discrete visual tokens predicted by the patch selector, along with the patch tokens selected based on a Keeping ratio of 0.25. We found that the patch selector tends to choose patches related to the semantics of discrete visual tokens, thereby supplementing the missing low-level fine-grained information of the discrete tokens. *The aforementioned findings further validate that multi-granularity hybrid encoding facilitates mutual assistance and synergy between discrete and continuous representations, thereby achieving efficient multi-granularity semantic encoding.*

**The impact of discrete visual tokens on multi-image reasoning**: To further validate the role of discrete visual tokens in the inference process of multi-image instructions, we visualize the attention weights of the last layer of the LLM. As illustrated in Figure 6, we have MaVEn compare the commonalities between two images. For inputs using only continuous visual tokens, we observe that the model's attention during inference is primarily focused on text tokens, disregarding visual tokens. This may still be due to the lower semantic granularity of continuous visual tokens, making it difficult to establish direct semantic associations. However, with inputs encoded using multi-granularity visual hybrid encoding, we notice that the model establishes attention associations with discrete visual tokens when answering questions. This indicates that, *in multi-image scenarios, discrete visual tokens guide the LLM to focus on visual information during decoding.*

## 5 Conclusion

In conclusion, we introduces MaVEn, a novel Multi-granularity Hybrid Visual Encoding framework designed to enhance multi-image reasoning in MLLMs. By combining discrete visual symbols for semantic abstraction with continuous sequences for detailed features, MaVEn improves both understanding and processing efficiency. Our dynamic reduction mechanism and multi-stage training strategy further enhance performance. Experimental results confirm that MaVEn significantly boosts MLLM capabilities in both multi-image and single-image contexts.

## 6 Acknowledgement

This work is supported by the National Natural Science Foundation of China (623B2007) and CCF-Zhipu Large Model Innovation Fund(NO.CCF-Zhipu202415).

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
