# OpenReview forum: "MaVEn: An Effective Multi-granularity Hybrid Visual Encoding Framework for Multimodal Large Language Model"
_NeurIPS.cc/2024/Conference — NeurIPS 2024 poster_

### Official Review · Reviewer_HJUS · 2024-07-09

**Soundness:** 3
**Presentation:** 3
**Contribution:** 3
**Rating:** 8
**Confidence:** 4

**Summary:**

The paper presents a novel approach, termed MaVEn, which aims to enhance the performance of MLLMs in multi-image scenario by integrating discrete visual symbol sequences with traditional continuous representation sequences. This dual strategy is designed to bridge the semantic discrepancies between visual and textual information. The approach also incorporates a dynamic reduction mechanism for long-sequence continuous features, aiming to boost processing efficiency in scenarios involving multiple images.

**Strengths:**

1) The manuscript makes a significant contribution by proposing a hybrid model that combines both discrete and continuous data representations. This is a promising approach to mitigate the issues of semantic gaps in multimodal learning. The dynamic reduction mechanism for handling long visual sequences is also an innovative solution that could have broad applications in the field.
2) The experiments conducted are robust and comprehensive,  as highlighted in section 4.4, is crucial in demonstrating the effectiveness of the visual hybrid encoding. This section effectively showcases how MaVEn performs under different scenarios, providing empirical evidence of its versatility and reliability.
3) The paper is generally well-written and organized. The methodology section is well-articulated and provides a clear explanation of how MaVEn operates.

**Weaknesses:**

To solidify the claims regarding the efficacy of the discrete visual symbol sequences used in MaVEn, it would be recommended to conduct experiments comparing the performance of these different discrete representation techniques, such as VQGAN[1] or VQVAE[2].
[1]Taming Transformers for High-Resolution Image Synthesis.
[2] Neural Discrete Representation Learning.

**Questions:**

The manuscript presents a well-structured and insightful study. I currently have no further questions.

**Limitations:**

yes.

---

> ### Author Rebuttal · Authors · 2024-08-06
>
> We would like to express our sincere gratitude for your insightful comments and the recognition of our work. We have carefully considered your suggestion and provide a detailed response below.
>
>
> ###  1.  it would be recommended to conduct experiments comparing the performance of these different discrete representation techniques, such as VQGAN or VQVAE.
>
> Thank you very much for your insightful suggestion. We fully acknowledge the importance of validating the efficacy of the discrete visual symbol sequences utilized in MaVEn by comparing them with other discrete representation techniques, such as VQGAN and VQVAE. These methods are indeed prominent in the field and have demonstrated significant success in various applications.
>
> In response to your recommendation, we have conducted additional experiments to compare the performance of MaVEn using different discrete representation techniques, including VQGAN and VQVAE. Additionally, we explored the potential of combining these techniques to further expand the vocabulary of the large language model (LLM).
>
> Below are the results of our comparative experiments:
>
> |   Visual Discrete Representation | Code Book Size | DemonBench Ave score | SEED Bench acc-video | VQA | MME | MMbench test |
> |-------|--------------------------------|----------------|----------------------|----------------------|-----| --------------|
> |  SEED                   |   8192     | 39.0           | 42.1                 | 79.1                 | 1530| 65.2| -            |
> |  VQGAN              |       1024   | 37.1           | 39.2                 | 77.3                 | 1441| 61.3| -            |
> |  VQVAE              |       1024  | 36.6           | 38.4                 | 76.3                 | 1380| 60.2| -            |
> |  SEED+VQGAN       |        9216      | 39.7           | 42.8                 | 79.5                 | 1521| 65.8| -            |
>
> Our conclusions are as follows:
> 1. Using SEED as the discrete visual token yields better performance compared to VQGAN and VQVAE.
> 2. Combining different discrete tokenizers can enhance the model's performance. We believe this improvement is due to the different visual semantic information encoded by the distinct codebooks. By integrating multiple codebooks, we achieve a richer and more comprehensive visual semantic representation, which in turn helps improve the model's overall performance.
>
> We appreciate your valuable feedback, which has significantly contributed to enhancing the robustness of our findings. We look forward to any further comments or suggestions you may have.
>
> Thank you for your thoughtful consideration.

---

> > ### Comment · Reviewer_HJUS · 2024-08-09
> > **Thank you very much for your reply.**
> >
> > My question has been resolved very well, and I am very satisfied with the experiments you added. Especially, the experiments you added have demonstrated that using multiple different discrete visual encodings simultaneously can further improve model performance, which is a very important discovery.  I'd like to try to follow this suit in my own work.
> > Additionally, I have read the comments from other reviewers as well as your reply, which has raised a new question for me. I noticed that in your reply, you mentioned that you believe the higher semantic granularity of discrete visual tokens helps MLLM better understand the semantic of images. But I also noticed that you used a self-regressive generation task for discrete image tokens in the second stage of training. Is it because you used this task that focused on discrete image tokens that it is the key to improving model performance? I would be very grateful if you could address this question and help me further improve my score.

---

> > > ### Author Response · Authors · 2024-08-09
> > > **Response to Reviewer HJUS**
> > >
> > > We sincerely appreciate your valuable suggestions and are honored by your recognition of our experiments. We will incorporate the aforementioned experiments and conclusions into the revised manuscript. Regarding the second issue you raised in your comments, we address your concerns from two perspectives:
> > >
> > > 1. **The semantic distribution of visual discrete tokens is closer to the text semantic distribution, aiding the alignment of image-text semantics by MLLM, which is the core reason for the model's performance improvement**: We encourage you to refer to Figure 6 in our paper and Figure 3 in the PDF provided in this rebuttal. We have conducted a visual analysis of the attention weight distribution of all features during the decoding phase of the LLM. Additionally, we clustered the discrete visual tokens. From the quantified experiments mentioned above, we observed that the semantics of visual discrete tokens are very close to the text semantics. Recent works [1][2][3] suggest that the most crucial step for multimodal large models is the alignment of image-text semantics, and visual representations based on discrete visual tokens are evidently easier to align with textual representations. Furthermore, the ablation study presented in Table 4 of our paper also supports this point where we found that the introduction of discrete tokens improved the model's performance on DemonBench from 30.66 to 39.51.
> > >
> > > 2. **We only used autoregressive image generation in the second phase of model training. The purpose of autoregressive image generation is solely to better aid the LLM in learning the embedding representation of visual discrete tokens, which is an essential step in introducing visual discrete representations**. During our training process, we only applied autoregressive image generation in the second phase of model training. Please refer to Figure 3 in our paper, where we show that only the embedding layer of the LLM was trained in this phase. Therefore, in our work, the use of autoregressive image generation is solely to better aid the LLM in learning the embedding representation of visual discrete tokens, helping the LLM understand visual discrete tokens. This is an essential step in introducing visual discrete representations. In fact, in previous experiments where we attempted to learn visual discrete tokens without autoregressive image generation, our results were as follows:
> > >
> > > | w/ Visual Discrete Tokens | w/ Image Regressive Generation | DemonBench Performance |
> > > |---------------------------|-------------------------------|------------------------|
> > > | ✓                         | ✓                             | 39.0                   |
> > > | ✓                         | ×                             | 32.14                  |
> > > | ×                         | ×                             | 30.66                  |
> > >
> > > We found that without autoregressive image generation, the embedding layer of the MLLM model struggled to effectively learn the embedding representation of visual discrete tokens, thereby limiting performance improvement on multiple images. Only by incorporating autoregressive image learning can the MLLM embedding layer better learn the embedding representation of discrete tokens and enhance performance.
> > >
> > > We hope our response adequately addresses your concerns.
> > >
> > > [1] Mind the Gap: Understanding the Modality Gap in Multi-modal Contrastive Representation Learning. NIPS 2022
> > >
> > > [2] InternVL: Scaling up Vision Foundation Models and Aligning for Generic Visual-Linguistic Tasks. CVPR2024
> > >
> > > [3] Visual Instruction Tuning. NIPS 2023

---

> > > > ### Comment · Reviewer_HJUS · 2024-08-12
> > > > **Thank you very much to the authors for the detailed response.**
> > > >
> > > > You have addressed my concerns, and I will increase my score.

---

> > > > > ### Author Response · Authors · 2024-08-12
> > > > >
> > > > > Thank you very much for your feedback and the time you dedicated to reviewing our paper. We greatly appreciate your insights and are pleased to hear that the clarifications we provided were helpful in addressing your concerns. If you have any further questions or suggestions, please do not hesitate to reach out. We would be happy to discuss them with you.

---

### Official Review · Reviewer_TdkD · 2024-07-12

**Soundness:** 2
**Presentation:** 3
**Contribution:** 3
**Rating:** 6
**Confidence:** 4

**Summary:**

This paper presents a Multi-granularity Visual Encoding framework (MaVEn) for better multi-image reasoning. MaVEn combines discrete visual symbols and continuous representation sequences, as well as designing a dynamic reduction mechanism to efficiently and effectively process and interpret information from multiple images. Experimental results demonstrate its effectiveness in various multi-image benchmarks.

**Strengths:**

1. The paper is well-structured and clear in its presentation.
2. I am highly impressed by the author's methodological design, particularly the Multi-Granularity Hybrid Encoding component, which I believe makes valuable and insightful contributions to the research community.
3. The thorough experiments and visualizations presented in the paper effectively demonstrate the efficacy of the proposed method.

**Weaknesses:**

1. In Stage 3, would the adjustments to the Visual Projector affect the performance of the Patch Selector, since that component has been frozen?
2. The steps of training the patch selector using Grounded SAM annotated data seem a bit redundant. Directly selecting patches based on the similarity between the patch and the discrete token may be a simpler and more effective approach.
3. The role of the continuous tokens has not been well validated. In Figure 6, the attention seems to barely focus on the continuous tokens. Does this suggest that the continuous tokens have little impact on the performance, and they could potentially be discarded? Is it possible that the current evaluation design is unable to fully reflect the role of the continuous tokens?
4. In line 202, '567' should be '576'.

**Questions:**

Overall, I appreciate this paper, but some of the concerns raised in the 'Weaknesses' part, especially Q3, have prevented me from giving a higher score.

**Limitations:**

None.

---

> ### Author Rebuttal · Authors · 2024-08-06
>
> ### 1. Would the adjustments to the Visual Projector affect the performance of the Patch Selector?
>
> The adjustments made to the Visual Projector in Stage 3 will not affect the performance of the Patch Selector. As shown in Figure 2(b) in our paper, the process is designed such that **we first use the Patch Selector to select the important patch tokens. Only after this selection has been made, we then input the selected patch tokens into the Visual Projector.** This sequential process ensures that the modifications made to the Visual Projector do not influence the operation of the Patch Selector.
>
> ### 2. Directly selecting patches based on the similarity between the patch and the discrete token may be a simpler and more effective approach.
>
> Thank you for your insightful suggestion.
>
> 1. We have indeed experimented with a similar approach where we directly select patches based on the attention weights between the <EOS> token in the discrete visual token sequence and all other continuous visual tokens. However, the results are shown below which were not satisfactory. This indicates that directly using attention weights may not be an effective method to measure the similarity between discrete tokens and continuous tokens.
>
> | Token Selection Method| DemonBench Ave Score| VQA | MMbench |
> | --------|----------|-------|-----|
>  |  Selection based on attention weights |  33.8 | 74.5     | 62.1        |
>  |  Selection with Patch Selector |  39.0 | 79.1  | 65.2  |
>
>
>
> 2. On the other hand, Grounded SAM is designed to locate relevant visual regions based on textual semantics, and given the fact (as shwon in the figure 3 in the rebuttal pdf) that for the paired image-text sample, the semantics encoded by the text captions is similar to semantics encoded by the visual discrete tokens, we decided to use Grounded SAM to generate pseudo-labels. This allows the patch selector to learn the similarity between visual discrete tokens and patches more effectively.
>
> Overall, we appreciate your suggestion and believe that the use of Grounded SAM to train the patch selector, although seemingly redundant, is a necessary step to ensure the effectiveness of our model.
>
>
> ### 3. The role of the continuous tokens has not been well validated.
>
> Your comments are greatly appreciated, and we will address your concerns in two parts:
>
> 1. **Clarification of Figure 6 and the importance of discrete Tokens**: We apologize for any confusion caused by Figure 6. In fact, the text question in the case presented in Figure 6 is "What is the similarity between Figure 1 and Figure 2?"  This case was  to highlight the challenges faced by recent MLLMs, particularly in multi-image contexts, **where the semantic granularity of the continuous visual representations and text tokens differ significantly.** In this specific case, the American flag represents a higher-dimensional, coarse-grained semantic entity that continuous visual tokens struggle to effectively encode alone. This results in the MLLMs paying less attention to continuous visual tokens during the decoding phase.  However, upon introducing discrete visual tokens, which align more closely in semantic granularity with text tokens, the MLLM was able to better grasp the high-level semantic  from the discrete tokens, thereby focusing more attention on these tokens and enhancing its ability to establish semantic associations across multiple images.
>
>    - To further verify the semantics of the discrete visual tokens that the MLLM focuses on, as illustrated in the figure 3 within the rebuttal PDF, we collected images that contain the discrete visual tokens targeted by the LLM. We discovered that these images consistently feature the American flag.
>
> 2. **The Role of Continuous Tokens-Encoding Fine-Grained Visual Details**: We are not suggesting that continuous visual tokens are useless. In fact, continuous visual tokens and discrete tokens are complementary and indispensable. Continuous tokens encode a amount of fine-grained semantic information in the image, so when the model faces scenarios that require understanding of fine-grained details in the image (where discrete visual tokens often do not contain this information), we actually need continuous visual tokens. To better validate this point, we conducted the following experiment:
>
>     - We randomly collected 500 images from the COCO dataset and tasked a GPT-4 model to generate questions about detailed object information in each image (e.g., asking about the shape, color, number, and size of certain objects) along with four different options and the correct answer. We then tested the performance on this dataset using MaVEN with only discrete visual tokens, only continuous visual tokens, and both, as shown below table, we found that the performance of the model using only discrete visual tokens was very poor, while the performance of the model using only continuous tokens was very close to that of the model using both, which also demonstrates the importance of continuous visual tokens for understanding detailed image information.
>
>     -  As shown in Fugure 1 in the submitted pdf in this rubuttal, we have also visualized the attention distribution of MaVEn for both discrete and continuous visual tokens when the model is asked about some details of the image (e.g.,"what is the color the dog in this image?"). We found that MLLMs also pay attention to continuous visual tokens. Which also suggests that the continuous visual tokens provide the fine-grained detial informations for our model.
>
>
> | Model | w/ Continuous token | w/ Discrete token | Accuracy |
> |-------|---------------------|-------------------|----------|
> | MaVEN | ✓       | ×                 |    35.2      |
> | MaVEN | ×       | ✓                 |      69.3    |
> | MaVEN | ✓       | ✓                 |      70.5    |
>
>
> ### 4. In line 202, '567' should be '576'.
>
> We appreciate your attention to detail and have corrected this error in the revised version of the paper.

---

> ### Author Response · Authors · 2024-08-12
>
> Dear Reviewer TdkD,
>
> First and foremost, please allow me to extend our deepest appreciation for the time and effort you have devoted to reviewing our paper.
>
> Moving forward, as the discussion phase is approaching its end, we are confident that we have comprehensively addressed the concerns you raised. We would greatly appreciate it if you could take a moment to review our responses. Your insights are important to us, and we are eager to hear your thoughts on the revisions we have made.
>
> Thank you once again for your attention and assistance.

---

> > ### Comment · Reviewer_TdkD · 2024-08-12
> >
> > Thanks for your response, which resolved most of my concerns. I would increase the score.

---

> > > ### Author Response · Authors · 2024-08-12
> > >
> > > We are truly grateful for the time and effort you invested in reviewing our paper and for your thoughtful feedback. Your insights have been invaluable, and we are glad that our clarifications effectively addressed your concerns.
> > >
> > > Best regard.

---

### Official Review · Reviewer_od3A · 2024-07-13

**Soundness:** 2
**Presentation:** 2
**Contribution:** 2
**Rating:** 6
**Confidence:** 2

**Summary:**

This paper introduces MaVEn, a Multi-granularity Visual Encoding framework that enhances Multimodal Large Language Models (MLLMs) in multi-image reasoning by combining discrete visual symbol sequences with traditional continuous representation sequences. Experimental results show that MaVEn significantly improves MLLMs' understanding in complex multi-image scenarios and boosts performance in single-image contexts.

**Strengths:**

1. The paper is well-organized, from problem, motivation, approach and experimental validation.

2. The proposed innovative multi-granularity approach includes 1) hybrid visual encoding and 2) dynamic reduction mechanism. The hybrid visual encoding captures both coarse-grained semantic concepts and fine-grained features, effectively bridging the semantic gap between visual and textual data.To enhance processing efficiency, a dynamic reduction mechanism is proposed, which selectively reduces long-sequence continuous features. This approach maintains essential information while reducing computational overhead.

3. The paper validates MaVEn’s effectiveness using several benchmarks, including DEMONBench and SEED-Bench, which encompass multi-image reasoning and video understanding tasks. MaVEn achieves superior performance compared to state-of-the-art models like LLaVA1.5, Otter, and others.Besides multi-image tasks, MaVEn also performs well in single-image benchmarks such as Visual Question Answering (VQA) and MMBench, showcasing its versatility.

**Weaknesses:**

see questions.

**Questions:**

I am not an expert in this domain. But I do have two questions for authors:

1. Why not directly operate on continuous features to reduce feature redundancy since the approach doesn't learn from coarse to fine, like gating?

2. It's not quite clear how discrete tokens help non-single image understanding. The obvious advantage of using discrete tokens is to improve efficiency only?

---

> ### Author Rebuttal · Authors · 2024-08-06
>
> Thank you for your recognition of our work and for your insightful questions.
> ### 1.  Why not directly operate on continuous features to reduce feature redundancy ?
>
> This is an excellent question. Directly reducing redundancy in continuous features can be approached in two main ways:
>
> 1. **Token Selection with global semantic:** This could be done by using the attention weights from the image global semantic token (e.g., <EOS> token)  to select important continuous visual tokens [1], [2].
> 2. **Merging Visual Tokens:** This can be done using latent queries (e.g., InstructBLIP[3]  used q-former to extract the fixed length visual continuous sequence) or convolutional network (e.g., QwenVL[4]) to merge long sequences of continuous visual tokens.
>
> We have experimented with these methods as per your suggestion, and the results are as follows:
>
> | Model | Method | DemonBench Ave Score| VQA | MMbench |
> |-------|--------|----------|-------|-----|
> |MaVEn | Token Selection with global semantic|  33.8 | 74.5     | 62.1        |
> | MaVEn | Token Merging based on Q-former | 29.4 | 71.1  | 54.2 |
> | MaVEn | Our Method |  39.0 | 79.1  | 65.2  |
>
> We found that these methods were less effective than our proposed approach, which combines discrete visual tokens with continuous tokens. From these results, we have the following inferences:
>
> 1. Compared with the first type of methods, our method is similar to them but is more efficient and accurate because our method utilizes coarse-grained discrete visual features and fine-grained continuous features to encode complementary information. In our method, discrete visual features capture high-level, coarse-grained information (e.g., "snowman," "American flag" as shown in Figure 6 in our paper), while continuous features capture fine-grained details of the image. In contrast, traditional methods based on global image semantic representations for token selection lack high-dimensional semantic guidance, resulting in lower accuracy in token selection.
>
> 2.  The second type of method, which merges visual information, compresses the data and loses important information, thereby impairing the MLLM's understanding of the image.
>
> To further illustrate the superiority of our method over traditional token selection methods, we conducted an experiment. We randomly selected 500 images from the COCO dataset and used Grounding SAM to segment relevant regions based on textual semantics of image caption as ground truth. We then measured the accuracy of selecting 20%, 40%, 60%, and 80% of tokens that hit the ground truth region for both approaches. As shown in below table, our method significantly outperformed the global visual semantic-based token selection method, demonstrating its effectiveness.
>
> | Token Selection Method |  20% | 40% | 60% | 80% |
> |------------------------|----------------|----------------|----------------|----------------|
> | Token Selection with global semantic | 93.4 | 90.5 | 84.4 | 79.4 |
> | Our Proposed Method | 74.8 | 66.3 | 61.1 | 52.3 |
>
>
>
> ### 2. It's not quite clear how discrete tokens help non-single image understanding. The obvious advantage of using discrete tokens is to improve efficiency only?
>
> Thank you for your insightful question.
>
> 1. To better address your concern, we would like to draw your attention to Figure 6 of our paper, which visualizes the Average Attention Weights with Only Continuous Visual Tokens and the Average Attention Weights with Multi-granularity Hybrid Visual Encoding. The text question in the case presented in Figure 6 is, "What is the similarity between Figure 1 and Figure 2?" This case highlights the challenges faced by recent MLLMs, particularly in multi-image contexts, where **the semantic granularity of continuous visual representations and text tokens differ significantly, making it difficult for MLLMs to understand and capture high-dimensional semantic information from images.** In this specific case, the American flag represents a higher-dimensional, coarse-grained semantic entity that continuous visual tokens alone struggle to effectively encode. As shown in the visualization of Average Attention Weights with Only Continuous Visual Tokens in Figure 6, this results in the MLLMs paying less attention to continuous visual tokens during the decoding phase.
>
> 2. **Upon introducing discrete visual tokens, which align more closely in semantic granularity with text tokens, the MLLM was able to better grasp the high-level semantics from the discrete tokens.** This alignment allowed the model to focus more attention on these tokens, thereby enhancing its ability to establish semantic associations across multiple images.
>
> 3. Moreover, to further verify the semantics of the discrete visual tokens that the MLLM focuses on, as illustrated in the figure within the rebuttal PDF, we collected images that contain the discrete visual tokens targeted by the LLM. We discovered that these images consistently feature the American flag. Therefore, it can be inferred that the semantics of the corresponding discrete visual tokens are associated with the American flag.
>
> **In summary, while discrete tokens do improve efficiency, their primary advantage lies in their ability to bridge the semantic gap between multi-image visual representations and text representations.** This alignment enhances the model's ability to understand high-level semantics and establish meaningful associations across multiple images, ultimately improving multi-image understanding.
>
> ### Reference
>
> [1] *An Image is Worth 1/2 Tokens After Layer 2: Plug-and-Play Inference Acceleration for Large Vision-Language Models.*
>
> [2] *Not All Patches are What You Need: Expediting Vision Transformers via Token Reorganizations.*
>
> [3] *InstructBLIP: Towards General-purpose Vision-Language Models with Instruction Tuning*
>
> [4] *Q*wen-VL: A Frontier Large Vision-Language Model with Versatile Abilities*

---

> ### Author Response · Authors · 2024-08-12
>
> Dear Reviewer od3A,
>
> Firstly, we would like to extend our sincere gratitude for the time and effort you have dedicated to reviewing our manuscript. Your insights and feedback are invaluable to us. As the discussion phase is nearing its conclusion, we believe we have addressed the concerns you raised in your review. We would be grateful if you could review our latest responses at your earliest convenience. Your further comments would be immensely helpful in refining our paper and moving forward in the review process.
>
> Thank you once again for your attention and assistance.

---

> > ### Comment · Reviewer_od3A · 2024-08-13
> >
> > thanks for the rebuttal! I am now leaning towards weak accept.

---

> > > ### Author Response · Authors · 2024-08-14
> > >
> > > Thank you for taking the time to read and consider our rebuttal. We greatly appreciate your positive feedback and are pleased to learn that you are now leaning towards a weak accept. We value your expertise and the insights you have provided throughout the review process. Your constructive comments have helped us identify areas for improvement and have contributed to enhancing the quality of our work.

---

### Official Review · Reviewer_9o1r · 2024-07-19

**Soundness:** 3
**Presentation:** 3
**Contribution:** 2
**Rating:** 4
**Confidence:** 4

**Summary:**

The paper introduces MaVEn, a framework designed to improve Multimodal Large Language Models (MLLMs) in understanding and reasoning across multiple images. Unlike current MLLMs, which are mainly focused on single-image interpretation, MaVEn integrates both coarse-grained semantic concepts and fine-grained details. This combination bridges the gap between visual and textual data, enhancing the model's ability to process multiple images. The framework also includes a mechanism for efficiently handling long sequences of features. Experiments show that MaVEn boosts performance in multi-image scenarios and also provides benefits for single-image tasks.

**Strengths:**

1. The concept is logical, and the paper is straightforward to read.
2. The concept of using both discrete and continuous visual tokens is intriguing.
3. I appreciate the author's use of figures 2 and 3, which help clarify the overall framework and training process.

**Weaknesses:**

1. In Tables 1, 2, and 3, the author does not compare some of the latest methods, such as mini-Gemini, MiniCPM, XComposer, and InternVL.
2. The paper primarily claims its main advantage is in multi-image tasks; however, the author only tests on DEMON and SEED benchmarks. Testing on additional multi-image benchmarks, such as MMBench-Video and MME-Video, would be more compelling.
3. Figure 6 is unclear. In the first figure, I see only two vertical lines on discrete tokens. Does this mean that only discrete tokens have attention weight?
4. The paper does not report the computational complexity compared to other methods.
5. Figure 5 shows that the selected tokens are mostly related to objects. Would it be beneficial to directly use Grounding Sam for token selection? A comparison might be interesting to see.
6. The training pipeline is complex and involves four stages. What is the training cost?

**Questions:**

Please refer to the weakness section.

---

> ### Author Rebuttal · Authors · 2024-08-06
>
> ### 1. Not compare some of the latest methods
> Thank you for your valuable comments.  We have reproduced the results of mini-Gemini, MiniCPM-V, and InternVL and compared MaVEn with them. It is important to emphasize that our experiments were based on LLaVA 1.5. To validate our method's effectiveness on the most current MLLMs, we further retrained MaVEn on the latest SOTA single-image MLLM LLaVA Next, we named it MaVEn-NEXT. The results are as follows:
>
> |Model|LLM|DemonBench Ave score|SEED Bench acc-video|VQA|MME|MMbench test|
> |-|-|-|-|-|-|-|
> |miniGemini|Vicuna 7b| 31.4 | 38.6| 65.2| 1523 |69.3 |
> |InternVL-Chat-v1| Vicuna 7b | 30.3| 40.3| 79.3| 1525 |74.7 |
> |MiniCPM-V v2.5| Llama3 8b |33.9 | 40.5| 80.3 |**1916** | **77.2**|
> |LLaVA 1.5| Vicuna 7b |30.6 | 37.3 |78.5 |  1511| 64.3 |
> |LLaVA Next| Llama3 8B|31.3|39.1 | 79.3 | 1591| 72.6 |
> |MaVEn| Vicuna 7b| 39.0| 42.1|79.1 |1530 |65.2 |
> |MaVEn-Next| Llama3 8B| **41.2** | **44.3** | **80.7** | 1623 | 75.5|
>
> We observed that MaVEn remains superior multi-image performance which demonstrating the effectiveness of our method. Moreover, MaVEn-NEXT shows improved single-image performance, reaching SOTA or comparable results.
> ### 2. Testing on additional multi-image benchmarks
> Thank you for your suggestion.
> Indeed, MMBench-Video and MME-Video were released after the NIPS submission deadline.  Moreover, based on your suggestion, we evaluated our model on MMBench-Video and MME-Video during the rebuttal period. During the evaluation, we extracted 8 frames from each video as input:
>
> |Model |CP | FP-S | FP-C| HL| LR| AR | RR| CSR| TR|
> |-|-|-|-|-|-|-|-|-|-|
> |InternVL-Chat-v1.5-[8f] |1.26 |**1.51** |1.22|**1.01**|**1.25**| **0.88** | 1.40 | **1.48**| 1.28 |
> |mPLUG-Owl2-[8f] |1.15| 1.34| 1.18 |0.99 |1.15| 0.63|1.33| 1.30| 1.03|
> |Qwen-VL-Chat-[8f] |0.52| 0.44| 0.62|0.33| 0.53| 0.45 |0.59| 0.50| 0.36|
> |Idefics2-8B-[8f]| 1.10| 1.23| 1.07| 0.89 |1.06 |0.77| 1.27| 1.41|1.11|
> |MaVEn-[8f] | **1.32** |1.32 |**1.24**| 0.96|1.18| 0.83 | **1.45**  |1.44| **1.31**|
>
> We found that MaVEn performs better than MLLM models such as mPLUG-owl2, Qwen-VL, Idefics2-8B, and its performance is comparable to that of InternVL-Chat-1.5.
>
> For the MME-Video results, please refer to Table 1 in the rebuttal pdf, where MaVEn also achieve competitive performance.
> ### 3. Figure 6 is unclear.
> 1. Clarification of Figure 6: We apologize for any confusion caused by Figure 6. In fact, the text question in the case is "What is the similarity between Figure 1 and Figure 2?" . **In this case, the American flag represents a coarse-grained semantic entity that continuous visual tokens struggle to effectively encode alone.** This results in the MLLMs paying less attention to continuous visual tokens during the decoding phase.
> 2. Importance of continuous visual tokens: When the model faces scenarios that require understanding of fine-grained details in the image, MLLMs actually need continuous visual tokens. To better validate this point, as shown in Fugure 1 in the submitted pdf in this rubuttal, we have visualized the attention weights distribution of MaVEn when the model is asked about the details of the image. We found that MLLMs also pay attention to continuous visual tokens.
> 3. Moreover, we conducted the following experiment: We randomly collected 500 images from the COCO dataset and tasked GPT4-o to generate questions about detailed object information in each image (e.g., asking about the shape, color, number, and size of an objects) . We then tested the accuracy on this dataset using MaVEN with only discrete visual tokens, only continuous visual tokens, and both, as shown in below table. We found the performance of the model using only discrete visual tokens was very poor.
>
> | Model | w/ Continuous token | w/ Discrete token | Accuracy |
> |-------|---------------------|-------------------|----------|
> | MaVEN | ✓       | ×                 |    35.2      |
> | MaVEN | ×       | ✓                 |      69.3    |
> | MaVEN | ✓       | ✓                 |      70.5    |
>
> ### 4. Report the computational complexity compared to other methods.
> In response, we have conducted a thorough evaluation:
> 1. Experiment Setting: Specifically, we set the input image size to 336x336 and the number of text input tokens to 24. We then measured the  throughput and FLOPs of 10 inference step for different numbers of images (ranging from 2 to 8) across several models: LLaVA 1.5, MaVEn, QwenVL, InternVL. The inference batch size is 1 and was performed on single 80G A100 GPU.
> 2. Experiment Results: As shown in Figure 2 of the submitted pdf file, we observed that as the number of images increases, MaVEn exhibits higher efficiency. This is primarily because MaVEn encodes a lower number of continuous visual tokens, which reduces the computational burden.
> ### 5.Directly use Grounding Sam for token selection?
> Very insightful question!
> 1. First, we want to highlight that Grounded SAM is engineered to segment image regions based on their corresponding textual information. However, if user instructions fail to provide explicit textual semantic cues for image segmentation (such as "What are the differences between image one and image two?"), it may lead to inaccurate region segmentation by Grounded SAM.
> 2. To fully address your concern, we tested the direct use of Grounded SAM for selecting image patches in both multi-image and single-image benchmarks. The results are as follows:
>
> | Method                 | DemonBench Ave Score| VQA | MMbench |
> |-------------------------|-|--------|---------|
> | MaVEn-Patch Selector    | 39.0 | 79.1     | 65.2        |
> | MaVEn-Grounding SAM     | 27.3 |   71.2   | 52.5        |
>
> The results indicate a significant decline when using Grounded SAM for selecting image patches, highlighting the effectiveness of patch selector.
> ### 6. training cost?
> We used 8*80G A100 GPU for training, which overall took about 122 hours. We give more detials in the table2 of the rebuttal pdf.

---

> ### Author Response · Authors · 2024-08-12
>
> Dear Reviewer 9o1r,
>
> We would like to sincerely thank you for the time and effort you have dedicated to reviewing our paper. Your insights and feedback have been invaluable in helping us improve our work.
>
> As the discussion phase is nearing its conclusion, we believe we have addressed your concerns in our recent replies and would greatly appreciate it if you could take a moment to review our responses. Your insights are important to us, and we are eager to hear your thoughts on the revisions we have made. Thank you once again for your time and consideration.

---

### Official Review · Reviewer_Rxew · 2024-08-11

**Soundness:** 3
**Presentation:** 3
**Contribution:** 3
**Rating:** 7
**Confidence:** 4

**Summary:**

This paper proposes MaVEn, a novel multi-granularity hybrid visual encoding framework for multimodal large language models (MLLMs). MaVEn aims to improve MLLMs' capabilities in multi-image reasoning by combining discrete and continuous visual representations. The authors design a dynamic reduction mechanism to reduce the computational overhead of long continuous sequences. Experimental results demonstrate that MaVEn significantly improves performance on both multi-image and single-image benchmarks.

**Strengths:**

This paper proposed a novel approach to combine the strength of both discrete and continuous visual representation as well as dynamic reduction mechanism.
In addition, the authors conduct comprehensive experiments on both multi-image and single-image benchmarks to demonstrate the effectiveness of MaVEn.

**Weaknesses:**

While the MaVEn proposed a novel approach to combine the advantage of discrete and continuous visual info, the system can be a bit over complex for serving / maintenance in real applications.

In addition, the paper didn't discuss the computation complexity of MaVEn and compare it with other existing models.

**Questions:**

1. how does MaVEn compare with other multi-granularity approaches?
2. Could you elaborate more on the computation complexity and efficiency?

**Limitations:**

Need more discussion on the model's limitation and computational complexity

---

> ### Author Response · Authors · 2024-08-12
>
> ## 1. The system can be a bit over complex for serving & compare computation complexity with other existing models.
> Thank you for your insightful comments. We appreciate your concern and would like to address it comprehensively.
> We acknowledge that the training process of our model might appear complex. However, we have conducted a thorough analysis comparing our model to recent SOTA models including QwenVL[1], InternVL[2], LLaVA1.5[3] in terms of training data size, training GPU count, inference FLOPs and latency .
> To evaluate the FLOPs and latency performance, we set the input image size of 336x336 and 24 text input tokens. We measured throughput and FLOPs over 10 inference steps using 4 input images, with an inference batch size of 1, on a single 80G A100 GPU.
>
> *Table 1. Comparision with different MLLMs on training data size, training GPU count, FLOPs, Latency and benchmark performance.*
> | Model | Training Data Size | Training GPU Count | Inference FLOPs | Inference Latency | DemonBench ave Score (Multi-image) | VQAv2 dev (Single Image) |
> |-------|--------------------|-----------|-----------------|-------------------|--------------------------|------------------------|
> | QwenVL 7B |1.5B | 640×A100(80G) | 212 | 2.22 |29.9 | 78.2|
> | InternVL 7B | 6B+ | Unkown| 340 | 1.53 | 30.3 |**79.3** |
> | LLaVA 1.5 7B |**1.2M** | **8×A100(80G)** | 193 | 2.46 | 30.6|78.5 |
> | MaVEN 7B | 7M | **8×A100(80G)**   | **163** | **2.64** | **39.0**|79.1 |
>
> As shown in below table 2 and table 3. We also report trends in FLOPs and latency across different numbers (from 2 to 8) of image inputs.
>
> *Table 2. FLOPs across different numbers of image inputs for various models.*
> | Image Nums | LLaVA | MaVEn | QwenVL | InternVL |
> |------------|-------|-------|--------|----------|
> | 2          | 104.44| 130.32| 120.33 | 210.33   |
> | 4          | 193.6 | 163.21| 212.04 | 340.53   |
> | 6          | 290.1 | 209.24| 324.23 | 480.2    |
> | 8          | 402.4 | 268.42| 450.22 | 670.21   |
>
> *Table 3. Latency across different numbers of image inputs for various models.*
> | Image Nums | LLaVA | MaVEn | QwenVL | InternVL |
> |------------|-------|-------|--------|----------|
> | 2          | 3.24  | 2.94  | 3.05   | 2.34     |
> | 4          | 2.46  | 2.64  | 2.22   | 1.53     |
> | 6          | 1.75  | 2.48  | 1.45   | 0.89     |
> | 8          | 1.21  | 2.22  | 0.88   | 0.33     |
>
> We have following conclusion:
> 1. The results in Table 1 demonstrate that our approach actually incurs lower overhead in both training and inference stages compared to these models. **This suggests that, contrary to initial impressions, our method is indeed practical and efficient for deployment and maintenance in real-world scenarios.**
> 2. The results in Tables 2 & 3, we observed that as the number of images increases, MaVEn exhibits higher efficiency. This is primarily because **MaVEn encodes a lower number of continuous visual tokens, which reduces the computational burden.**
>
> We hope this detailed analysis alleviates your concerns about the complexity of our model in practical applications.
>  ## 2. How does MaVEn compare with other multi-granularity approaches?
> In response to your recommendation, we have conducted additional experiments to compare the performance of MaVEn using different multi-granularity  techniques, where we utilize VQGAN and VQVAE to replace the SEED tokenizer. Additionally, we explored the potential of combining these techniques.
> Below are the results of our comparative experiments:
>
> | Visual Discrete Representation | Code Book Size | DemonBench Ave score | SEED Bench acc-video | VQA  | MME  | MMbench test |
> |-------------------------------|----------------|----------------------|----------------------|------|------|--------------|
> | SEED                          | 8192           | 39.0                 | 42.1                 | 79.1 | 1530 | 65.2         |
> | VQGAN                         | 1024           | 37.1                 | 39.2                 | 77.3 | 1441 | 61.3         |
> | VQVAE                         | 1024           | 36.6                 | 38.4                 | 76.3 | 1380 | 60.2         |
> | SEED+VQGAN                    | 9216           | 39.7                 | 42.8                 | 79.5 | 1521 | 65.8         |
>
> Our conclusions are as follows:
> 1. Using SEED as the discrete visual token yields better performance compared to VQGAN and VQVAE.
> 2. Combining different discrete tokenizers can enhance the model's performance. We believe this improvement is due to the different visual semantic information encoded by the distinct codebooks. By integrating multiple codebooks, we achieve a richer and more comprehensive visual semantic representation, which in turn helps improve the model's overall performance.
>
> [1] Qwen-VL: A Frontier Large Vision-Language Model with Versatile Abilities.
>
> [2] InternVL: Scaling up Vision Foundation Models and Aligning for Generic Visual-Linguistic Tasks.
>
> [3] Improved Baselines with Visual Instruction Tuning.

---

> ### Author Response · Authors · 2024-08-12
>
> Dear Reviewer Rxew,
>
> Thank you very much for taking the time and effort to review our paper. We sincerely appreciate your valuable feedback and insights.
>
> We wanted to inform you that we have provided our responses in the comments section. We would greatly appreciate it if you could take a moment to review our replies.
>
> Thank you once again for your time and consideration.

---

> ### Comment · Reviewer_Rxew · 2024-08-13
>
> Thanks for the rebuttal! The comparison looks nice, I changed the rating to 7 (Accept)

---

> > ### Author Response · Authors · 2024-08-14
> >
> > We sincerely appreciate you taking the time to read our rebuttal and for your positive feedback. We are thrilled to learn that our clarifications and the additional comparison have addressed your concerns effectively. Thank you for your thorough evaluation and for your support. We deeply value your expertise and the insights you have provided throughout the review process.

---

### Author Rebuttal · Authors · 2024-08-06

We would like to express our sincere gratitude for your thorough and insightful reviews of our manuscript. We greatly appreciate the time and effort you have invested in providing valuable feedback and suggestions, which will undoubtedly help us improve the quality and clarity of our work.

We are pleased to note that the reviewers have acknowledged several strengths of our paper.  Reviewer **9o1r** found *"The concept of using both discrete and continuous visual tokens is intriguing."* Reviewer **od3A** commended our paper *"the innovative multi-granularity approach"*. Reviewer **TdkD** and Reviewer **HJUS** commended *"our work makes a valuable and insightful contribution to research community."* Reivewers **od3A, TdkD** and **HJUS**  think *our experiments effectively validates the effectiveness the proposed method.*

We are encouraged by these positive comments and will strive to address the concerns and suggestions raised by the reviewers to further enhance our manuscript.

Moreover, we would like to kindly remind you that we have included the visualized experimental data and certain tabular data within the PDF document submitted for this rebuttal phase. We encourage you to review these materials. Your attention to these details is greatly appreciated.

---

> ### Author Response · Authors · 2024-08-09
> **Gentle Reminder:   Review Our Response and Engage in Further Discussion**
>
> Dear Reviewers,
>
> We would like to kindly remind you to review our response to your comments and concerns. We have made our best efforts to address all the points raised by the reviewers and have provided detailed explanations and additional experiments to support our work.
> The valuable insights and suggestions provided by the reviewers have helped us to strengthen our contributions and clarify the key aspects of our research. We are eager to engage in further discussion with the reviewers and welcome any additional feedback or questions you may have. Your expertise and comments are highly appreciated.
>
> Thank you once again for your time and consideration.

---

### Decision · Program_Chairs · 2024-09-25

**Decision:**

Accept (poster)

**Comment:**

The reviewers unanimously agree that the proposed approach is both novel and compelling, with some highlighting its significant technical contributions and broad applicability across various scenarios (HJUS, TdkD). The reviewers also acknowledge the thoroughness of the experiments and the clarity of the presentation. Initial concerns were raised regarding the computational cost (9o1r, Rxew), the effectiveness of continuous visual tokens (TdkD), and the adequacy of comparisons with other methods. However, the authors have addressed these concerns comprehensively in their rebuttal. After consideration of the reviews and the authors' responses, the Area Chair finds no remaining critical issues with the paper. The Area Chair agrees with the reviewers on the value of the proposed approach and recommends acceptance of the paper.